# Synergizing Liquid Biopsy and Hybrid PET Imaging for Prognostic Assessment in Prostate Cancer: A Focus Review

**DOI:** 10.3390/biom15071041

**Published:** 2025-07-18

**Authors:** Federica Stracuzzi, Sara Dall’ Armellina, Gayane Aghakhanyan, Salvatore C. Fanni, Giacomo Aringhieri, Lorenzo Faggioni, Emanuele Neri, Duccio Volterrani, Dania Cioni

**Affiliations:** 1Nuclear Medicine Unit, Azienda Ospedaliera di Rilievo Nazionale e di Alta Specializzazione (ARNAS) Garibaldi, 95124 Catania, Italy; federicastra@hotmail.it; 2Nuclear Medicine Unit, Service Area Department, Azienda Socio-Sanitaria Territoriale (ASST)-Rhodense P.O. Bollate, 994301 Milan, Italy; sdallarmellina@asst-rhodense.it; 3Nuclear Medicine Unit, Department of Translational Research and of New Surgical and Medical Technology, University of Pisa, 56126 Pisa, Italy; duccio.volterrani@unipi.it; 4Academic Radiology, Department of Translational Research and of New Surgical and Medical Technology, University of Pisa, 56126 Pisa, Italy; fannisalvatoreclaudio@gmail.com (S.C.F.); giacomo.aringhieri@unipi.it (G.A.); lorenzo.faggioni@unipi.it (L.F.); emanuele.neri@unipi.it (E.N.); 5Academic Radiology, Department of Surgical, Medical, Molecular Pathology and Emergency Medicine, University of Pisa, 56126 Pisa, Italy; dania.cioni@unipi.it

**Keywords:** cancer biomarkers, cell free DNA, circulating tumor cells, imaging biomarkers, liquid biopsy, PET/CT, PET/MR, precision medicine, prostate cancer, translational oncology

## Abstract

Positron emission tomography (PET) and liquid biopsy have independently transformed prostate cancer management. This review explores the complementary roles of PET imaging and liquid biopsy in prostate cancer, focusing on their combined diagnostic, monitoring, and prognostic potential. A systematic search of PubMed, Scopus, and Cochrane Library databases was conducted to identify human studies published in English up to January 2025. Seventeen studies met the inclusion criteria and were analyzed according to PRISMA guidelines. Across the included studies, PET-derived imaging metrics, such as metabolic activity and radiotracer uptake, correlated consistently with liquid biopsy biomarkers, including circulating tumor cells and cell-free DNA. Their joint application demonstrated added value in early detection, treatment monitoring, and outcome prediction, particularly in castration-resistant prostate cancer. Independent and synergistic prognostic value was noted for both modalities, including survival outcomes such as overall survival and progression-free survival. Combining PET imaging and liquid biopsy emerges as a promising, non-invasive strategy for improving prostate cancer diagnosis, monitoring, and therapeutic stratification. While preliminary findings are encouraging, large-scale prospective studies are essential to validate their integrated clinical utility.

## 1. Introduction

Prostate cancer (PCa) is the most prevalent malignancy among adult males worldwide and remains one of the leading causes of cancer-related mortality [1]. The clinical management of PCa has evolved significantly over recent years, driven by substantial advances in diagnostic and prognostic technologies. In particular, innovations in molecular imaging and biomarker-based approaches have opened new avenues for early detection, treatment stratification, and disease monitoring [2].

Among these, positron emission tomography/computed tomography (PET/CT) with prostate-specific membrane antigen (PSMA) ligands has gained prominence for its high sensitivity and specificity in identifying tumor burden, especially in advanced stages. PSMA-targeted imaging has increasingly influenced clinical decision-making by enabling the accurate localization and characterization of disease extent. A prospective multicenter trial by Hofman et al. demonstrated that PSMA PET-CT significantly outperforms conventional imaging for both sensitivity (92% vs. 65%) and specificity (95% vs. 74%) in the initial staging of high-risk PCa, influencing treatment intent in 28% of patients [3].

Concurrently, liquid biopsy techniques, including the analysis of circulating cell-free DNA (cfDNA), circulating tumor DNA (ctDNA), and circulating tumor cells (CTCs), have emerged as powerful, minimally invasive alternatives to traditional tissue biopsies. Additionally, exosomes in bodily fluids carry tumor-specific molecular information, making them promising biomarkers for prostate cancer. Urinary liquid biopsy and serum biomarkers also contribute to advancing PCa detection and personalized treatment. Together, these tools hold great potential for improving diagnosis, prognosis, and therapy. These approaches allow for the real-time monitoring of tumor-specific genetic and molecular alterations, providing dynamic insights into tumor evolution and resistance mechanisms [4]. In metastatic castration-resistant prostate cancer (mCRPC), ctDNA levels have been shown to correlate with tumor burden and predict clinical outcomes independently of PSA, as reported in a landmark study by Wyatt et al. [5].

The rationale for integrating PET imaging with liquid biopsy lies in the complementary strengths of the two modalities. While PET imaging offers spatial and metabolic characterization of disease, liquid biopsy captures molecular-level changes in real time. Emerging data suggest that combining PSMA PET imaging with ctDNA profiling may enable the early detection of therapeutic resistance and residual disease, as shown in recent integrative analyses by Graff et al. and Sartor O. [6,7]. Emerging evidence highlights a strong correlation between PSMA PET/CT-derived parameters such as SUVmax and ctDNA levels, suggesting that tumors with high tracer uptake are more likely to shed detectable ctDNA into circulation [8].

Together, these approaches represent a synergistic strategy that could enhance disease staging, prognostication, and treatment personalization.

This review aims to comprehensively assess the interplay between liquid biopsy and molecular imaging in PCa. In doing so, we also evaluate the association between PET-derived parameters and liquid biopsy biomarkers, with particular attention to cfDNA and ctDNA levels across different stages of disease, including hormone-sensitive and castration-resistant PCa. Furthermore, we explore the independent prognostic value of liquid biopsy markers and PET imaging metrics, such as PSMA-derived tumor volume (PSMA-TV), metabolic tumor volume (MTV), and standardized uptake value (SUVmax), in predicting overall survival (OS) and progression-free survival (PFS). Finally, this review aims to identify key research gaps and outline future directions for integrating liquid biopsy and hybrid PET imaging into clinical practice, with the overarching goal of advancing precision monitoring and personalized treatment strategies in PCa.

## 2. Materials and Methods

A systematic literature search was conducted using the PubMed, Scopus, and Cochrane Library databases to identify studies evaluating the use of nuclear imaging and liquid biopsy biomarkers in PCa. The search included peer-reviewed articles published in English up to January 2025, limited to human subjects. The following keywords and their Boolean combinations (AND, OR, NOT) were used: “liquid biopsy”, “circulating DNA”, “prostate cancer”, “PET/CT”, “PET/MR”, and “molecular imaging.”

Two independent reviewers (nuclear medicine physicians) screened the titles and abstracts of all retrieved records to exclude ineligible studies. Discrepancies were resolved through consensus. Full texts of potentially relevant studies were assessed in detail. A manual cross-reference check of selected studies was also performed to identify additional relevant articles.

The eligibility criteria included original studies in humans reporting both PET imaging parameters (e.g., SUVmax, PSMA-TV, MTV) and liquid biopsy biomarkers (e.g., cfDNA, ctDNA, CTCs) in PCa patients. Studies were excluded if they were reviews, editorials, letters, preclinical studies, case reports, or not published in English.

For each included study, we extracted the following data: authors, publication year, journal, country of origin, sample size, clinical context, study design, primary endpoints, and, where available, PET and liquid biopsy parameters.

Out of 121 full-text articles retrieved, 27 duplicates were removed, leaving 94 articles for screening. Of these, 34 were review articles, 17 off-topic, 7 editorials or letters, 1 a case report, 8 preclinical studies, and 1 not in English. An additional 8 studies were excluded due to insufficient PET or liquid biopsy data. During full-text screening, we identified two studies by Kluge et al., reporting results from the same patient cohort enrolled between March 2019 and August 2021 at the Medical University of Vienna. To avoid duplicate data inclusion, we retained the article with the larger sample size (*n* = 148) and broader outcome analysis [9], and excluded the overlapping study [10] (*n* = 130). Ultimately, 17 studies met the inclusion criteria and were included in the qualitative synthesis. The study selection process adhered to PRISMA (Preferred Reporting Items for Systematic Reviews and Meta-Analyses) guidelines and is illustrated in the PRISMA flow diagram (Figure 1).

No formal risk of bias assessment was conducted, as this review aimed to synthesize clinical findings rather than perform a meta-analysis.

## 3. Results

### 3.1. Study Characteristics

The 17 included studies involved a total of 1197 patients across various disease stages, from localized to metastatic castration-resistant PCa (mCRPC). Study designs were predominantly prospective (*n* = 8), retrospective (*n* = 6), or combined (*n* = 3). Imaging modalities included [68Ga]Ga-PSMA-11 PET/CT, [18F]FCH PET/CT, [18F]PSMA-1007 PET/CT, and [18F]NaF PET/CT. Liquid biopsy markers assessed included cfDNA, ctDNA, ptDNA, CTCs, exosomes, AR-V7, and other RNA-based molecular profiles. Detailed characteristics are summarized in Table 1. Figure 2 provides a visual overview of the timeline of included studies and the distribution of PET radiotracers used.

### 3.2. PET Imaging Findings

PET imaging consistently demonstrated high sensitivity in detecting tumor burden across PCa subtypes. PSMA-PET enabled superior total disease consolidation compared to conventional imaging alone, such as in the study by Phillips et al. [13]. Several studies showed that imaging-derived metrics, such as SUVmax, MTV, and PSMA-TV, correlated with clinical parameters. Emmett et al. used PSMA and FDG PET/CT together with ctDNA and CTC profiling to derive biomarkers predictive of treatment response and prognosis [18]. Kluge et al. reported that PSMA-TV was highly prognostic across hormone-sensitive and castration-resistant stages and showed added value when interpreted alongside ctDNA concentrations [9]. Similarly, Conteduca et al. demonstrated a strong association between SUVmax, MTV (from [18F]FCH PET), and plasma tumor DNA (ptDNA), reinforcing the synergistic role of liquid biopsy and PET imaging [19]. Heterogeneity in PET response, especially SUVhetero, was identified by Kyriakopoulos et al. as a predictor of PSA progression [15].

### 3.3. Liquid Biopsy Biomarkers

The liquid biopsy results demonstrated dynamic changes during treatment and were associated with tumor burden. cfDNA concentrations increased after chemotherapy in the study by Kwee et al. [11], while MYC amplification was detected via cfDNA in all PET-positive patients in the study by Aggarwal et al. [12]. CTC screening provided a highly sensitive biomarker for the early detection of cancer, with higher CTC counts being associated with higher risk of malignancy [14]. CTCs were commonly used in early detection, with studies by Ried et al. reporting high sensitivity and specificity (PPV 99%, NPV 97%) [16]. AR-V7 and AR-FL mRNA expression correlated with tumor load and treatment response in mCRPC [17]. Novel approaches, such as extracellular vesicle digital scoring [21] and Glycan Score assays [26], provided promising results for stratifying localized vs. metastatic disease. Modlin et al. reported that 27 gene signatures for PCa are concordant with tissue mRNA levels, indicating that measuring blood expression can provide a minimally invasive genomic tool that may facilitate PCa management [24].

### 3.4. Correlations Between PET and Liquid Biopsy

Multiple studies reported significant associations between PET metrics and liquid biopsy markers. Kwee et al. found a negative correlation between cfDNA and tumor activity (r = −0.50, *p* = 0.01) [11]. Conteduca et al. observed that higher ptDNA levels corresponded to greater MTV and SUVmax [19]. Kluge et al. found that PSMA-TV correlated with cfDNA in advanced disease, but not in hormone-sensitive PCa [9]. In a combined analysis, both PSMA-TV and ctDNA were independently associated with survival in multivariate models.

### 3.5. Complementary Diagnostic Value

Studies supported the potential for combining PET and liquid biopsy to enhance early diagnosis. Ghous et al. demonstrated that the integration of [68Ga]Ga-PSMA-11 PET/CT with CTCs and exosomes provided 66.7% sensitivity and 93.3% specificity in detecting PCa [22]. In Derlin et al., CTC-based PSMA profiling showed spatial correlation with PET-derived PSMA phenotypes [20]. Similarly, the digital scoring assay by Wang et al. distinguished metastatic disease from localized disease even when lesions were not visible on imaging [21].

### 3.6. Prognostic Implications

PET-derived parameters such as PSMA-TV, MTV, and SUVmax were repeatedly associated with survival outcomes. Conteduca et al. and Kessel et al. linked MTV to both PFS and OS, while Kluge et al. demonstrated that PSMA-TV was a stronger predictor of OS than cfDNA alone [9,17,19] and that both PET and ctDNA metrics were found to be independently prognostic for survival across disease stages. Gupta et al. further showed that CTC profiling could identify patients likely to respond to PSMA-targeted radioligand therapy (RLT) [25].

## 4. Discussion

This review confirms that the integration of hybrid PET imaging and liquid biopsy represents a promising paradigm for precision medicine in PCa. While each modality has demonstrated diagnostic and prognostic value independently, their combination enables a more comprehensive assessment of disease burden, biology, and treatment response, especially in castration-resistant disease [9,19,20].

PET imaging, particularly with PSMA-targeted tracers, offers unparalleled spatial resolution and quantification of tumor volume and heterogeneity [27]. Meanwhile, liquid biopsy captures molecular and genetic changes in real time, often anticipating phenotypic shifts or emerging resistance mechanisms [15,17]. This dual approach addresses a critical gap in PCa management: the need to monitor both visible tumor burden and its molecular evolution dynamically and non-invasively.

A key insight from the reviewed studies is the strong correlation between PET-derived metrics (e.g., PSMA-TV, MTV, SUVmax) and liquid biomarkers (e.g., ctDNA, CTCs, ptDNA), supporting their mutual validation and reinforcing their combined prognostic utility. Notably, several authors demonstrated that high values of both imaging and molecular markers are associated with significantly worse OS and PFS [9,19]. Conversely, low tumor volume on PET combined with low ctDNA levels identified patients with excellent outcomes, highlighting the value of integrated risk stratification.

Beyond risk prediction, this synergy also reveals tumor heterogeneity and treatment resistance. Liquid biopsy detected AR-V7 expression and neuroendocrine differentiation even in patients with low or absent PSMA uptake on PET, suggesting early phenotypic divergence invisible to imaging alone [17,26]. Similarly, PSMA-negative CTCs correlated with poor response to radioligand therapy [20] while dynamic changes in CTCs during therapy anticipated therapeutic benefit [23].

Moreover, emerging evidence suggests that the combination of hybrid PET imaging and liquid biopsy not only improves diagnostic accuracy, but also enhances personalized therapeutic decision-making [28,29]. Recent studies have shown that integrating these modalities facilitates the early detection of minimal residual disease and guides timely modifications in treatment regimens, which is critical in managing aggressive and treatment-resistant prostate cancer phenotypes [3,30]. For instance, PET imaging can localize metastatic sites that may be missed by conventional imaging, while liquid biopsy provides insights into clonal evolution and emerging mutations that drive resistance to androgen receptor signaling inhibitors or chemotherapy [31,32]. This integrative approach also supports the development of novel theranostic strategies, such as targeted radionuclide therapies tailored to molecular profiles identified through ctDNA or CTC analysis [33]. Furthermore, longitudinal monitoring through liquid biopsy paired with serial PET scans offers a dynamic framework to assess treatment efficacy and detect early relapse, ultimately improving clinical outcomes [30,31]. Collectively, these advancements underscore the pivotal role of combining molecular and imaging biomarkers to refine risk stratification, optimize therapy selection, and monitor disease trajectory in prostate cancer management. In addition to established biomarkers, microRNAs (miRNAs) are important in prostate cancer development and treatment resistance. For example, miRNA-21 contributes to castration resistance, while miRNAs from the miRNA-200 and miRNA-17 families are linked to PSA response and better survival in CRPC patients treated with docetaxel. The miRNA-200 family regulates epithelial-to-mesenchymal transition, a key process in drug resistance and metastasis, and the miRNA-17 family affects immune response. These miRNAs could be useful biomarkers for monitoring treatment and predicting response.

### 4.1. Clinical Trials Ongoing

In recent years, the combined use of PET imaging and liquid biopsy has been gaining increasing importance, not only in clinical practice, but also within clinical trials, particularly in the context of PSMA-RLT. TAYLOR, a prospective, monocentric study (ClinicalTrials.gov ID: NCT06917781), is investigating the association between CTCs, [18F]DCFPyL PET/CT findings, and clinical outcomes in PCa patients. Patients are stratified by tumor burden, and the presence of CTCs is evaluated within each group. For those treated with [177Lu]PSMA-617, additional blood samples and clinical/imaging data are collected over time to assess treatment response and the potential prognostic value of CTCs. The ANGELA trial (ClinicalTrials.gov ID: NCT05188911) [34] is a prospective observational study designed to investigate the integration of molecular imaging, using both PSMA and FDG imaging and ctDNA, to assess lesion-level heterogeneity and monitor genomic changes during treatment with novel hormonal agents (Abiraterone, Prednisone, and Androgen deprivation therapy) in mCRPC patients. A prospective single-center one-arm Phase II clinical trial (ClinicalTrials.gov ID: NCT06220188) is currently enrolling PCa patients with confirmed biochemical recurrence (BCR) in the absence of radiologically detectable local recurrence after curative-intent primary treatment. Eligible patients receive two cycles of [177Lu]-Lu-PSMA-I&T. Among the exploratory outcomes assessed over an 18-month period are the quantification of ctDNA, enumeration of CTCs before and after treatment, and the analysis of molecular alterations in liquid biopsy markers induced by the therapy. This approach aims to provide insights into the biological response and potential predictive biomarkers associated with PSMA-RLT. Another prospective single-center one-arm Phase II clinical trial (ClinicalTrials.gov ID: NCT06259123) evaluates the impact of systemic radioligand therapy with [177Lu]-Lu-PSMA-I&T in patients with oligometastatic PCa scheduled for radical prostatectomy. The study investigates the role of neoadjuvant PSMA-RLT in modulating PSA kinetics, imaging response, pathological outcomes, and long-term oncological efficacy. Exploratory endpoints assessed over a 24-month follow-up include the enumeration of CTCs and the characterization of molecular alterations in both liquid biopsy markers and tumor tissue samples collected after PSMA-RLT, post-surgery, and during follow-up. These analyses aim to enhance the understanding of treatment-induced biological changes and to identify potential biomarkers predictive of response in the setting of early systemic intervention. A prospective phase II/III study (ClinicalTrials.gov ID: NCT03824275) will enroll 300 men with a diagnosis of PCa, who will undergo standard of care imaging (CT or MRI of the chest, abdomen, and pelvis, and [99mTc] bone scans) alongside routine laboratory assessments and [18F]DCFPyL PET/CT. Among the exploratory endpoints, evaluated over a 3.5-year follow-up, are the molecular characterization of ctDNA and exosomes, as well as the analysis of their correlation with disease burden. The ENZA-p study (ClinicalTrials.gov ID: NCT04419402) [35] is a phase II, multicenter, randomized clinical trial evaluating the safety and efficacy of adding [177Lu]Lu-PSMA to enzalutamide in patients with chemotherapy-naïve mCRPC. A total of 160 participants will be randomized 1:1 to receive enzalutamide alone or in combination with Lu-PSMA, with stratification based on disease burden (with [68Ga]PSMA PET/CT), prior systemic therapies, and study site. As part of the translational research program, liquid biopsy, including ctDNA and CTCs, will be collected at key time points (baseline, Day 92, and progression) to identify prognostic and predictive biomarkers associated with treatment response, resistance, and safety. Follow-up will continue until study completion, approximately 4 years from recruitment. EVO, a single-institution, single-arm, open-label Phase 2 trial (ClinicalTrials.gov ID: NCT06836726), investigates the efficacy of combining hypoxia-targeted therapy with androgen receptor signaling inhibitors (ARSIs) in patients with CRPC progressing after first-line ARSI. Following baseline dual-tracer PET/CT imaging with PSMA and FDG, patients receive the investigational treatment. Treatment response and disease progression are assessed every 8 weeks using whole-body PSMA PET/CT. In parallel, liquid biopsy samples, including cfDNA, PSA levels, neuroendocrine markers, and organ function tests are collected at each treatment cycle to monitor biological response and identify molecular correlates of efficacy and resistance. Assessments continue every 8 weeks until death.

### 4.2. Challenges

Despite these promising findings, several limitations warrant consideration. Many of the included studies were retrospective or involved small patient cohorts, limiting statistical power and generalizability. Heterogeneity in methodologies—including liquid biopsy platforms and PET quantification metrics—hinders direct comparisons and reproducibility [36]. Additionally, short follow-up durations in some studies may preclude definitive conclusions on long-term prognostic value. Furthermore, while ctDNA and CTCs have shown consistent value, cfDNA yielded variable prognostic correlations [9,15], suggesting the need for more robust standardization and analytical validation.

However, a major limitation remains the lack of harmonization among the various methodological approaches: each research group adopts different techniques for both the acquisition and interpretation of PET images and for the molecular data obtained from liquid biopsy, making cross-study comparison challenging.

To address this issue, international initiatives such as the European Association of Nuclear Medicine (EANM) Research 4 Life template (EARL), which focuses on standardizing PET procedures [37,38], and Blood Profiling Atlas in Cancer (BloodPAC) [39], which works on the qualification and harmonization of circulating biomarkers, are laying the groundwork for greater consistency and comparability of data.

### 4.3. Comparative Added Value of PET and Liquid Biopsy

Compared to conventional imaging techniques such as MRI, CT, or bone scintigraphy, PET imaging, particularly with PSMA-labeled tracers, offers superior sensitivity and specificity for detecting prostate cancer lesions, especially in biochemical recurrence or advanced disease [3,28]. Liquid biopsy, on the other hand, provides real-time molecular insights that conventional imaging cannot capture, such as the detection of AR-V7 expression or emerging resistance mutations [29].

While direct comparative studies are currently lacking, the integration of PET and liquid biopsy potentially outperforms each modality alone or in combination with traditional tools. Unlike MRI or CT, this dual approach may enable the earlier detection of micro-metastatic or molecular progression, guide patient selection for PSMA-targeted therapies, and identify resistance mechanisms not visible on imaging. Such advantages suggest a paradigm shift in the personalized management of prostate cancer, particularly in cases where imaging and histopathology alone may be insufficient.

Future comparative studies should evaluate the diagnostic and prognostic performance of PET plus liquid biopsy versus MRI, CT, and serum PSA, in well-defined clinical contexts such as initial staging, biochemical recurrence, or treatment monitoring. Until then, the integrated use of these two advanced modalities may complement, but not replace, standard tools, offering a more nuanced and dynamic assessment of disease.

### 4.4. Bridging Clinical Value with Real-World Applications

Beyond clinical potential, the integration of PET imaging and liquid biopsy into routine workflows also depends on real-world feasibility. Key factors include the availability of infrastructure, reimbursement frameworks, and the degree of standardization across imaging and molecular platforms. In many healthcare settings, logistical challenges, such as limited access to hybrid PET scanners or high-complexity molecular diagnostics, may constrain widespread adoption. Nevertheless, both modalities are increasingly represented in clinical trial protocols and incorporated into specialist guidelines, reflecting their growing perceived value and momentum toward broader implementation.

### 4.5. Future Directions

The combined use of PSMA PET parameters and liquid biopsy markers offers significant advantages over the use of either approach alone. While PET imaging provides spatial and functional information about tumor burden, liquid biopsy captures molecular and genetic alterations in real time. Integrating these complementary data improves the early detection of castration-resistant prostate cancer and enhances the prediction of overall and progression-free survival. The studies reporting correlations between PET and liquid biopsy markers support this synergistic value, demonstrating a more comprehensive and accurate assessment of disease status when both modalities are combined. Innovative biomarkers such as the Glycan Score and Met Score further enhance the role of liquid biopsy, offering higher specificity than PSA and correlating with PSMA-PET metrics in both localized and metastatic settings [21,26]. Their integration with PET data could support earlier detection of micro-metastatic disease and improve timing for therapeutic interventions.

Below is a summary of the two liquid biopsy scores discussed:•Glycan Score: Offers higher specificity than PSA; correlates with PSMA-PET parameters.•Met Score: Enhances detection sensitivity; correlates with PSMA-PET in both localized and advanced disease.

In parallel, the field is being transformed by the emergence of novel multiomics technologies, which further expand the possibilities of biological patient profiling. Approaches including transcriptomics and other system-level molecular analyses are opening new avenues for precision medicine, supporting the development of increasingly refined and integrated predictive models. There is also growing interest in integrating image-based data from radiomics into the multiomics framework, which aims to combine biomolecular information with imaging features to enhance disease characterization. Radiomics involves extracting quantitative features from medical images and correlating them with clinical outcomes, thereby enabling more precise and personalized diagnosis and prognosis. In prostate cancer (PCa), common imaging modalities include MRI, transrectal ultrasound, conventional CT, cone-beam CT, and molecular imaging techniques, particularly PET/CT with radiotracers such as radiolabeled PSMA and fluorine-18-labeled choline. This integration of radiomics with molecular imaging holds great promise for improving clinical decision-making by linking imaging phenotypes with underlying tumor biology [40].

## 5. Conclusions

This review demonstrates that the integration of liquid biopsy and PET imaging moves PCa management toward a multidimensional, personalized model. While PET imaging contributes spatial and functional information about tumor burden and heterogeneity, liquid biopsy offers molecular insights that can be captured non-invasively and longitudinally. Together, these modalities complement each other and enable a more dynamic, personalized understanding of disease status and treatment response. This approach is particularly valuable in capturing tumor complexity and treatment resistance in real time. Future prospective trials should aim to validate these findings across disease stages, ideally using standardized acquisition and quantification protocols. If confirmed, the use of combined imaging and molecular markers could guide treatment selection, monitor therapeutic efficacy, and ultimately improve outcomes for patients with PCa.

## Figures and Tables

**Figure 1 biomolecules-15-01041-f001:**
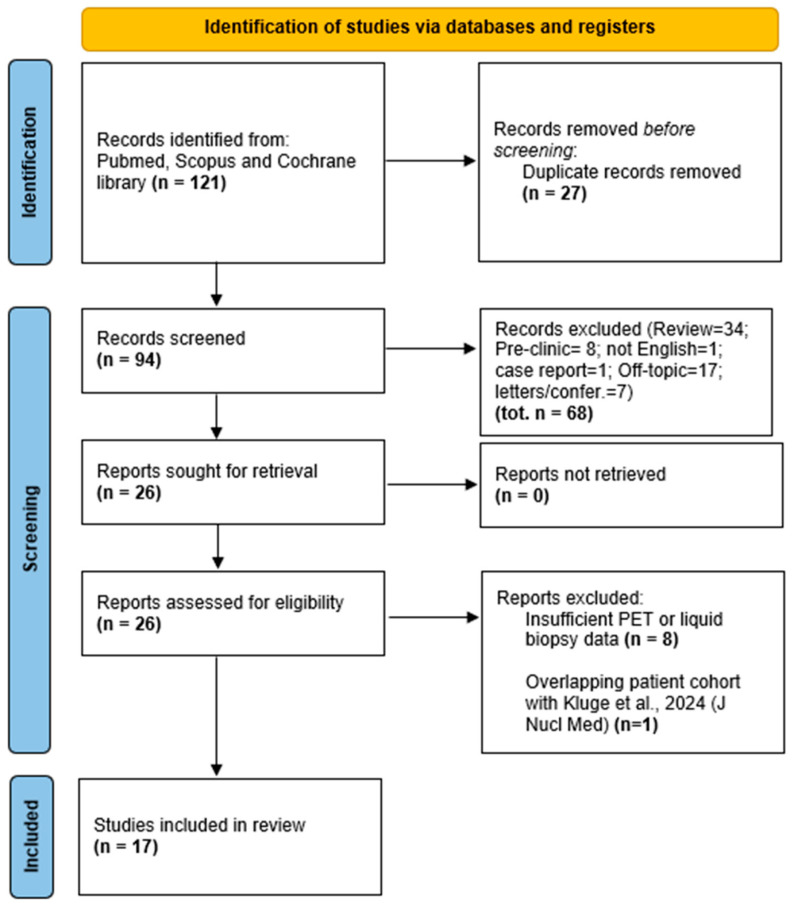
PRISMA flow diagram of study selection. The flowchart illustrates the process of study identification, screening, eligibility assessment, and inclusion according to PRISMA (Preferred Reporting Items for Systematic Reviews and Meta-Analyses) guidelines. A total of 121 records were retrieved through database searches. After removing duplicates and applying inclusion/exclusion criteria, 17 studies were included in the final qualitative synthesis.

**Figure 2 biomolecules-15-01041-f002:**
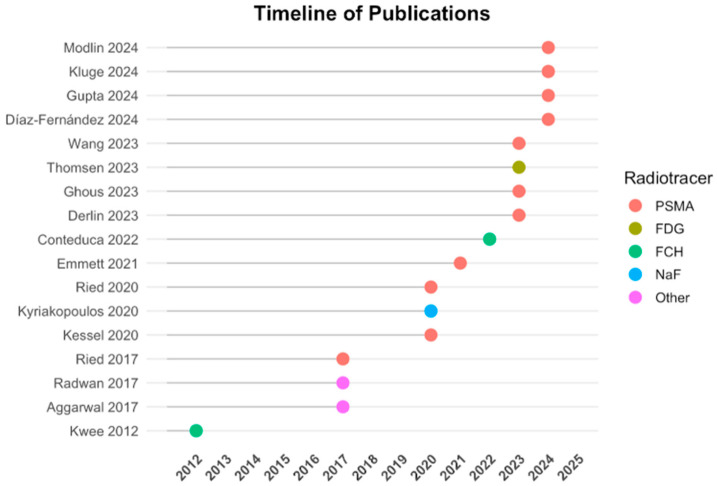
Timeline of publications and radiotracers used. Included studies are plotted by year of publication along the vertical axis. The color-coded dots indicate the primary PET radiotracer employed in each study: PSMA-based (red), FDG (olive), FCH (green), NaF (blue), and other tracers (magenta). This timeline highlights the recent predominance of PSMA-targeted PET imaging in PCa research and its integration with liquid biopsy methodologies.

**Table 1 biomolecules-15-01041-t001:** Summary of studies evaluating PET imaging and liquid biopsy biomarkers for diagnosis, prognosis, and treatment monitoring in prostate cancer.

Author, Year	Study Design	Patients (n)	Clinical Context	Imaging Modality	Liquid Biopsy Markers	Aim	Key Findings
Kwee S. et al., 2012 [11]	P	8	Metastatic castration-resistant PCa	[18F]FCH PET/TC	cfDNA	Evaluate cfDNA in response to chemotherapy	Significant correlation (r = −0.50, *p* = 0.01) between cfDNA levels and tumor activity on PET/CT
Aggarwal R. et al., 2017 [12]	P	18	Metastatic castration-resistant PCa	[68Ga]citrate-PET/TC and PET/MR	cfDNA	Detect MYC-positive prostate tumors	MYC gain detected in cfDNA for all PET-positive patients
Philips R. et al., 2017 [13]	CT	54	Oligo-metastatic PCa	[18F]DCFPyL PET	ctDNA	Evaluate PSA and imaging-based progression at 6 months post-SABR	SABR improved clinical outcomes; PSMA-PET enabled superior total disease consolidation compared to conventional imaging alone.
Ried K. et al., 2017 [14]	R	69	Localized disease/biochemical recurrence	[68Ga]Ga-PSMA-PET/CT	CTCs	Assess the relationship between CTC counts and cancer risk/status in patients undergoing treatment or asymptomatic screening.	Higher CTC counts strongly correlated with increased cancer risk; CTC screening demonstrated high sensitivity for early cancer detection.
Kyriakopoulos C.E. et al., 2020 [15]	CT	23	Metastatic castration-resistant PCa	[18F]NaF PET/TC	CTCs	Inter-lesional response heterogeneity in bone metastases via quantitative PET	SUVhetero changes were the strongest predictors of PSA progression; SUV-total increased at progression despite initial improvement during enzalutamide treatment.
Ried K. et al., 2020 [16]	R	20	Localized disease	PSMA-PET/TC	CTCs	Evaluate the ISET^®^-CTC test combined with prostate-specific markers.	The combination of ISET^®^-CTC and ICC-PSA-marker testing has a positive predictive value (PPV) of 99% and a negative predictive value (NPV) of 97%.
Kessel K. et al., 2020 [17]	P	19	Metastatic castration-resistant PCa	[18F]PSMA-1007 PET/TC	CTCs, AR-FL, AR-V7 mRNA	Correlate several clinical and molecular parameters with response to PSMA	AR-FL and AR-V7 might serve as prognostic biomarkers displaying high tumor burden in mCRPC patients prior to PSMA-RLT.
Emmett L. et al., 2021 [18]	CT	160	Metastatic castration-resistant PCa	[68Ga]Ga-PSMA and FDG PET/CT	cfDNA, CTCs	Identification of prognostic and predictive biomarkers from PSMA and FDG PET/CT and ctDNA.	Developed predictive and prognostic biomarkers to better guide treatment decisions.
Conteduca V. et al., 2022 [19]	P	102	Metastatic castration-resistant PCa	FCH-PET/CT	cfDNA, ptDNA	Investigate whether pretreatment ptDNA reflects metabolic tumor burden in combination with functional imaging.	A significant association was seen between ptDNA and SUVmax, MTV and TLA.
Derlin T. et al., 2023 [20]	P	20	Metastatic castration-resistant PCa	[68Ga]Ga-PSMA-11 PET/CT	CTCs	Explore the interrelation between CTCs and solid metastatic lesions	Liquid biopsy is complementary to PET for individual PSMA phenotyping of mCRPC.
Wang J.J. et al., 2023 [21]	R	40	Localized disease/metastatic PCa	[68Ga]Ga-PSMA-11 PET/CT, bone scan	mRNA, AR-V7, antiPSMA	Developing a PCa extracellular vesicle (EV) digital scoring assay (DSA) for detecting metastasis of PCa.	Met score distinguishes metastatic from localized PCa and reflects clinical behavior even when the disease was undetectable by imaging.
Ghous M.H. et al., 2023 [22]	R	55	Suspected or confirmed prostate cancer	[68Ga]Ga-PSMA-11 PET/CT	CTCs, ctDNA, exosomes	Find the role of liquid biopsy and molecular imaging for early diagnosis of PCa.	Liquid biopsy and molecular imaging have the potential to complement conventional screening methods for early PCa diagnosis.
Thomsen L.C.V. et al., 2023 [23]	C	18	Metastatic castration-resistant PCa	[18F]FDG PET/CT, 99Tc-bone scan	CTCs	Determined the safety and tolerability of cryoablation without and with checkpoint inhibitors.	Post-treatment symptoms were associated with CTCs presence while CTCs responses correlated with clinical outcomes; cryoimmunotherapy in mCRPC is safe and well tolerated.
Modlin I.M. et al., 2024 [24]	P	178	Localized disease/metastatic PCa	[68Ga]Ga-PSMA-11 PET/CT	CTCs	Development of a molecular assay from mRNA databases using machine learning in PCa.	Measuring blood expression provides a minimally invasive genomic tool that may facilitate PCa management.
Kluge K. et al., 2024 [9]	P/R	148	Metastatic hormone sensitive/metastatic castration resistant	[68Ga]Ga-PSMA-11 PET/CT	cfDNA	Evaluate the relationship and prognostic value of cfDNA and PSMA-TV in men with PCa	cfDNA does not reliably reflect total tumor burden or prognosis; although PSMA PET/CT provides a highly prognostic assessment of tumor burden across the spectrum of PCa disease progression.
Gupta S. et al., 2024 [25]	P	24	Metastatic castration-resistant PCa	[68Ga]Ga-PSMA-11 PET/CT	CTCs	CTCs may identify the patients most likely to benefit from PSMA-targeted RLT.	It demonstrates the potential to detect PSMA protein expression in CTCs from patients with mCRPC.
Diaz-Fernández A. et al., 2024 [26]	R	30	Localized and metastatic PCa	[18F]PSMA-1007 PET/TC	Glycan score	Using Glycan Score in cancer patients’ serum and proved its facility for stratification of PCa.	The Glycan Score test has a huge potential for accurate diagnosis and staging of PCa.

Abbreviations: PCa: prostate cancer; cfDNA: cell-free-DNA; PSA: prostatic specific antigen, SABR: stereotactic ablative body radiotherapy; PSMA: prostate-specific membrane antigen; CTC: circulating tumor cell; PET: positron emission tomography; TC: computed tomography; MR: magnetic resonance; FCH: fluoroCholine; DCFPyL: Piflufolastat; SUV: standard uptake volume; PVV: positive predictive value; NPV: negative predictive value; mCRPC: metastatic castration-resistant prostate cancer; MTV: molecular tumor volume; TLA: total lesion activity; NaF: sodium fluoride; ptDNA: plasma tumor-DNA; EV: extracellular vesicle; DSA: digital scoring assay; mRNA: messenger-RNA.

## Data Availability

No new data were created or analyzed in this study. All data analyzed during this study are available from the original publications cited in the manuscript.

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
