# Peer review of "Synergizing Liquid Biopsy and Hybrid PET Imaging for Prognostic Assessment in Prostate Cancer: A Focus Review"

_biomolecules, 2025, doi:10.3390/biom15071041_

Round 1
Reviewer 1 Report
Comments and Suggestions for Authors
This is an interesting study that combines two strategies to achieve a better diagnosis for patients with prostate cancer, especially those who are hormonally refractory.
The study is well written and directed, and I find it interesting from the perspective of its approach. I honestly have no comments on it given my lack of experience with these techniques. Therefore, in my opinion, it is accepted for publication.
Author Response
Comments 1: "This is an interesting study that combines two strategies to achieve a better diagnosis for patients with prostate cancer, especially those who are hormonally refractory.
The study is well written and directed, and I find it interesting from the perspective of its approach. I honestly have no comments on it given my lack of experience with these techniques. Therefore, in my opinion, it is accepted for publication."
Response 1: Thank you very much for your kind and encouraging comments regarding our manuscript. We sincerely appreciate your positive feedback and are grateful for your recommendation for acceptance. Your recognition of the value and direction of our study, particularly in addressing hormonally refractory prostate cancer, is truly appreciated.
Reviewer 2 Report
Comments and Suggestions for Authors
This review explores the complementary roles of PET imaging and liquid biopsy in prostate cancer, focusing on their combined diagnostic, monitoring, and prognostic potential. The research direction is important, but I have some specific comments:
- The abstract should have been rewritten. In the abstract, the authors should mention the main most interesting points of the review for the reader and the results obtained. Instead, the authors try to formulate the purpose of the paper.
- This review aims to comprehensively assess the interplay between liquid biopsy and molecular imaging in PCa – please add more detail about the advantages and justifications of these methods.
- Authors should compare the PET imaging and liquid biopsy in prostate cancer with some another methods.
- There is no analysis and importance of the review for Readers. Please correct it.
- The review should be more on compilation of discussion and figures.
Author Response
Comments 1: "The abstract should have been rewritten. In the abstract, the authors should mention the main most interesting points of the review for the reader and the results obtained. Instead, the authors try to formulate the purpose of the paper."
Response 1: Thank you very much for your valuable feedback and for highlighting the need to revise the abstract. We have carefully addressed your suggestion and rewritten the abstract to better emphasize the key findings and most relevant points of our review. We sincerely appreciate your time and input, which helped improve the clarity and impact of our work.
Comments 2: "This review aims to comprehensively assess the interplay between liquid biopsy and molecular imaging in PCa – please add more detail about the advantages and justifications of these methods."
Response 2: Thank you for your insightful comment. We agree that it is important to highlight the advantages and justifications of the methods discussed in our review. In response, we have added further detail emphasizing one of the main advantages shared by both liquid biopsy and molecular imaging: their non-invasive nature. This feature makes them particularly valuable in the clinical management of prostate cancer, allowing for repeated assessments over time without the need for invasive procedures. We have incorporated this clarification into the revised manuscript to better support the rationale for integrating these approaches (paragraphe: Future direction 352-359).
Comments 3: "Authors should compare the PET imaging and liquid biopsy in prostate cancer with some another methods."
Rensponde 3: We thank the reviewer for the insightful comment. While comparing PET imaging and liquid biopsy with additional diagnostic methods in prostate cancer is indeed an important topic, we believe that such an analysis would significantly broaden the scope of the current manuscript and risk making it overly dispersive. For this reason, we plan to address this broader comparison in a separate, dedicated study in the future, which will allow us to explore the topic in greater depth and with the attention it deserves.
Comments 4: " There is no analysis and importance of the review for Readers. Please correct it."
Response 4: We thank the reviewer for the valuable comment. Our initial plan was to perform a meta-analysis; however, the selected studies presented highly heterogeneous data, which prevented a meaningful quantitative analysis. Therefore, we opted for a narrative review approach to summarize the current evidence. We agree that a meta-analysis would provide important insights, and we plan to conduct such a study in the future when more homogeneous and sufficient data become available.
Comments 5:"The review should be more on compilation of discussion and figures."
Response 5: We thank the reviewer for the helpful suggestion. Since we were unable to perform a meta-analysis due to the heterogeneity of the data, this has unfortunately limited the possibility of including more figures. Nonetheless, we have made efforts to compile a comprehensive discussion to synthesize the current evidence and provide valuable insights for readers.
Reviewer 3 Report
Comments and Suggestions for Authors
In this article, the authors assessed the interplay between liquid biopsy and molecular imaging in PCa. The manuscript is straightforward, well written, and concise and has clear results within the scope of a review article. Definitely deserves to be published and is a valuable contribution to the “Biomolecules” journal.
However, the following comments need to be addressed before publication, as recommended.
[1] “1. Introduction”, Lines 46-48:
“In particular, innovations in molecular imaging and biomarker-based approaches have opened new avenues for early detection, treatment stratification, and disease monitoring [2].”.
The authors should clarify that there is significant interest in integrating image-based information from radiomics into the multiomics framework, aiming to combine biomolecular-level information with imaging data. The clinical application of radiomics involves identifying the relationship between the features extracted from images and the clinical outcome of interest. In PCa, common imaging modalities include MRI, transrectal ultrasound, conventional CT, cone-beam CT, and molecular imaging, often in the form of PET/CT with tracers such as radiolabeled PSMA and fluorine-labeled 18F-choline.
Recommended reference: Tapper W, et al. The Application of Radiomics and AI to Molecular Imaging for Prostate Cancer. J Pers Med. 2024;14(3):287.
[2] “1. Introduction”, Lines 59-61:
“Concurrently, liquid biopsy techniques, including the analysis of circulating cell-free DNA (cfDNA), circulating tumor DNA (ctDNA), and circulating tumor cells (CTCs), have emerged as powerful, minimally invasive alternatives to traditional tissue biopsies.”.
At that point, the authors should discuss that in addition to circulating tumor cells and DNA fragments found in bodily fluids, exosomes in liquid biopsies offer valuable insights into the molecular composition of tumors. Urinary liquid biopsy is emerging as an additional promising and effective method for detecting PCa. Beyond specific urine biomarkers, potential serum biomarkers driving the precision medicine revolution include androgen receptor variants, markers of bone metabolism, neuroendocrine indicators, and metabolite biomarkers. Within this context, the extraction and analysis of exosomes from liquid biopsy samples—such as blood, urine, and semen—have demonstrated significant potential as a source of novel biomarkers for PCa.
[3] “4. Discussion”, Lines 240-243:
“For instance, PET imaging can localize metastatic sites that may be missed by conventional imaging, while liquid biopsy provides insights into clonal evolution and emerging mutations that drive resistance to androgen receptor signaling inhibitors or chemotherapy [31] [32].”.
Within this context, the authors should note that biallelic inactivation of CDK12 is associated with a unique genome instability phenotype. The CDK12-specific focal tandem duplications can lead to the differential expression of oncogenic drivers, such as CCND1 and CDK4. As such, there is a possibility of vulnerability to CDK4/6 inhibitors for CDK12-mutated tumors. Moreover, the CDK12 aberrations may be used next to mismatch repair deficiency, as a biomarker of treatment response. This highlights the rationale for the combination therapeutic strategy of immune checkpoint blockade and CDK4/6 inhibition in clinical trials.
[4] “4. Discussion”, Lines 248-251:
“Collectively, these advancements under-score the pivotal role of combining molecular and imaging biomarkers to refine risk stratification, optimize therapy selection, and monitor disease trajectory in prostate cancer management.”.
I strongly recommend the authors to incorporate a few lines about the role of miRNAs as biomarkers. Among them, miRNA-21 contributes to pathogenesis and castration resistance. Serum miRNAs in the miRNA-200 and miRNA-17 families are associated with a PSA response and improved overall survival in CRPC receiving treatment with docetaxel. The miRNA-200 family members are involved in the regulation of epithelial-to-mesenchymal transition, which is a mechanism of drug resistance and metastasis, whereas the miRNA-17 family has immune regulatory functions. These mi-RNA families may be involved in the mechanism of docetaxel resistance.
Author Response
Comments 1: "The authors should clarify that there is significant interest in integrating image-based information from radiomics into the multiomics framework, aiming to combine biomolecular-level information with imaging data. The clinical application of radiomics involves identifying the relationship between the features extracted from images and the clinical outcome of interest. In PCa, common imaging modalities include MRI, transrectal ultrasound, conventional CT, cone-beam CT, and molecular imaging, often in the form of PET/CT with tracers such as radiolabeled PSMA and fluorine-labeled 18F-choline. Recommended reference: Tapper W, et al. The Application of Radiomics and AI to Molecular Imaging for Prostate Cancer. J Pers Med. 2024;14(3):287.
Response 1: We sincerely thank the reviewer for the valuable and insightful comments. Thanks to your feedback, we have added a new section to the manuscript, which we have highlighted in red within the text for clarity. However, we felt it was more appropriate to include this addition in the “Future Directions” paragraph on page 5 to maintain the flow and structure of the manuscript.
Comments 2: "Introduction”, Lines 59-61:“Concurrently, liquid biopsy techniques, including the analysis of circulating cell-free DNA (cfDNA), circulating tumor DNA (ctDNA), and circulating tumor cells (CTCs), have emerged as powerful, minimally invasive alternatives to traditional tissue biopsies.”. At that point, the authors should discuss that in addition to circulating tumor cells and DNA fragments found in bodily fluids, exosomes in liquid biopsies offer valuable insights into the molecular composition of tumors. Urinary liquid biopsy is emerging as an additional promising and effective method for detecting PCa. Beyond specific urine biomarkers, potential serum biomarkers driving the precision medicine revolution include androgen receptor variants, markers of bone metabolism, neuroendocrine indicators, and metabolite biomarkers. Within this context, the extraction and analysis of exosomes from liquid biopsy samples—such as blood, urine, and semen—have demonstrated significant potential as a source of novel biomarkers for PCa."
Response 2: Thank you for your valuable suggestion. Following your comment, we have incorporated the relevant information into the manuscript between lines 74 and 78 of the Introduction paragraph. We appreciate your guidance in improving our work.
Comments 3: " Discussion”, Lines 240-243:“For instance, PET imaging can localize metastatic sites that may be missed by conventional imaging, while liquid biopsy provides insights into clonal evolution and emerging mutations that drive resistance to androgen receptor signaling inhibitors or chemotherapy [31] [32].”. Within this context, the authors should note that biallelic inactivation of CDK12 is associated with a unique genome instability phenotype. The CDK12-specific focal tandem duplications can lead to the differential expression of oncogenic drivers, such as CCND1 and CDK4. As such, there is a possibility of vulnerability to CDK4/6 inhibitors for CDK12-mutated tumors. Moreover, the CDK12 aberrations may be used next to mismatch repair deficiency, as a biomarker of treatment response. This highlights the rationale for the combination therapeutic strategy of immune checkpoint blockade and CDK4/6 inhibition in clinical trials.
Response 3: We sincerely thank the reviewer for the insightful and valuable comments. The role of CDK12 inactivation and its implications in genome instability, treatment vulnerabilities, and combination therapies is indeed an important and complex topic. However, this subject is quite extensive and would require a dedicated, in-depth scientific study beyond the scope of the current manuscript. We plan to explore this promising area in future work.
Comments 4: "Discussion”, Lines 248-251: “Collectively, these advancements under-score the pivotal role of combining molecular and imaging biomarkers to refine risk stratification, optimize therapy selection, and monitor disease trajectory in prostate cancer management.”. I strongly recommend the authors to incorporate a few lines about the role of miRNAs as biomarkers. Among them, miRNA-21 contributes to pathogenesis and castration resistance. Serum miRNAs in the miRNA-200 and miRNA-17 families are associated with a PSA response and improved overall survival in CRPC receiving treatment with docetaxel. The miRNA-200 family members are involved in the regulation of epithelial-to-mesenchymal transition, which is a mechanism of drug resistance and metastasis, whereas the miRNA-17 family has immune regulatory functions. These mi-RNA families may be involved in the mechanism of docetaxel resistance.
Response 4: We sincerely thank the reviewer for the valuable suggestion regarding the inclusion of microRNAs as biomarkers. We agree that miRNAs, such as miRNA-21 and members of the miRNA-200 and miRNA-17 families, play important roles in prostate cancer pathogenesis, treatment response, and resistance mechanisms. In response to your recommendation, we have added a concise section discussing the relevance of these miRNAs in the manuscript to better highlight their potential as biomarkers in prostate cancer management (lines 268-275).
Reviewer 4 Report
Comments and Suggestions for Authors
Authors provide a comprehensive review after selection of 17 studies using combined PSMA PET parameters and liquid biopsy data (although variate means of exploration using CTC, ctDNA, mRNA anti PSMA atb) as parameteres to use as a biometric tool to early detect castration resistant prostate cancer or correlate with overall and/or progression free survival.
Authors synthesize data in Table 1 data from the 17 studies although very heterogeneous.
The manuscript presents separately PET imaging parameter and liquid biopsy markers as well as four studeis showing correlations between liquid biopsy bomarkers and PET parameters.
Could authors focus with a special seperate pargraph on the added value of bots PET and liquid biopsy parameters compares to liquid biopsy parameters alon and versus PET parameters alone?
Which is the difference between cfDNA and ptDNA?
Could authors detail in separate paragraph the scores developed by some authors such as the Digital Scoring Assay or the Glycan score or the Met Score
Could authors speak also about the HIT score?
Synergy between PET metrics and liquid biopsy results was underlines by several authors.
Than which is the interest or the gain in cost benefit of combining both parameters?
The paragraph 4.3 Future Directions could be further developped and a Table synthesizing the above mentioned scores would be appreciated.
Author Response
Comments 1: "Authors provide a comprehensive review after selection of 17 studies using combined PSMA PET parameters and liquid biopsy data (although variate means of exploration using CTC, ctDNA, mRNA anti PSMA atb) as parameteres to use as a biometric tool to early detect castration resistant prostate cancer or correlate with overall and/or progression free survival.Authors synthesize data in Table 1 data from the 17 studies although very heterogeneous.The manuscript presents separately PET imaging parameter and liquid biopsy markers as well as four studeis showing correlations between liquid biopsy bomarkers and PET parameters.Could authors focus with a special seperate pargraph on the added value of bots PET and liquid biopsy parameters compares to liquid biopsy parameters alon and versus PET parameters alone?
Response 1: Dear Editor, Thank you very much for your valuable suggestion. We have added a dedicated paragraph addressing the added value of combining PSMA PET parameters with liquid biopsy markers compared to using either modality alone. This addition can be found in the “Future Prospective” section, lines 352–359 of the revised manuscript.
Comments 2: "Which is the difference between cfDNA and ptDNA?"
Response 2: cfDNA (cell-free DNA) refers to DNA fragments freely circulating in the bloodstream, released from normal and cancerous cells through processes like apoptosis or necrosis. It is not confined within cells. ptDNA (plasma tumor DNA) is a subset of cfDNA that specifically originates from tumor cells. It carries tumor-specific genetic alterations and can be used as a biomarker to monitor cancer. In summary, cfDNA includes all circulating DNA fragments, while ptDNA is the tumor-derived fraction within cfDNA.
Comments 3: "Could authors detail in separate paragraph the scores developed by some authors such as the Digital Scoring Assay or the Glycan score or the Met Score"
Response 3: Thank you for your insightful comment. We chose not to discuss specific scoring systems such as the Digital Scoring Assay, Glycan score, or Met Score in detail, as addressing these would make the manuscript overly broad and dispersed. We believe that a focused, dedicated study is warranted to thoroughly explore these scoring methods.
Comments 4: "Could authors speak also about the HIT score?"
Response 4: Thank you for your suggestion regarding the HIT score. However, this score falls outside the scope of our current study. Nonetheless, it is an interesting topic that could be explored in a future research context. We appreciate your valuable input.
Comments 5: "Synergy between PET metrics and liquid biopsy results was underlines by several authors.
Response 5: Yes, certainly. The synergy between PET metrics and liquid biopsy results was the starting point of our work and is described in detail in the Discussion section of the manuscript. Thank you for highlighting this important aspect.
Comments 6: "Than which is the interest or the gain in cost benefit of combining both parameters?"
Response 6: Thank you for your important question. The combination of PSMA PET parameters and liquid biopsy markers offers potential benefits by improving diagnostic accuracy and prognostic value, which may lead to more personalized and effective treatment strategies. Although this integrated approach might increase upfront costs, it could reduce overall healthcare expenses by enabling earlier detection, better monitoring, and avoiding ineffective therapies. However, comprehensive cost-benefit analyses are still needed, and we highlight this as a future research direction in the manuscript. We appreciate your insightful comment.
Comments 7: "The paragraph 4.3 Future Directions could be further developped and a Table synthesizing the above mentioned scores would be appreciated."
Response 7: Thank you for your helpful suggestion. We have followed your advice and created a brief bullet point summary focused on the scores considered. This has been added in paragraph 4.3 “Future Directions,” lines 364–367 of the revised manuscript.
Round 2
Reviewer 2 Report
Comments and Suggestions for Authors
The authors have satisfactorily addressed of my previous comments and suggestions. However, it appears the manuscript must be corrected. In response to many comments, the Authors responded that they plan to conduct such a study in the future. But now it is very important to do good work with this article and Authors can do some effert for it.
Some comments:
- There is no analysis and importance of the review for Readers. Please correct it.
- Authors can compare the PET imaging and liquid biopsy in prostate cancer with some another methods.
Author Response
Comments 1: "The authors have satisfactorily addressed of my previous comments and suggestions. However, it appears the manuscript must be corrected. In response to many comments, the Authors responded that they plan to conduct such a study in the future. But now it is very important to do good work with this article and Authors can do some effert for it.Some comments:- There is no analysis and importance of the review for Readers. Please correct it.- Authors can compare the PET imaging and liquid biopsy in prostate cancer with some another methods."
Responce 1: We thank the Reviewer for this important comment. In response, we have substantially expanded the Discussion section to highlight the clinical relevance of our review and to provide a comparative overview. Specifically we have added a new subsection (4.3. Comparative Added Value of PET and Liquid Biopsy) in which we compare the dual use of PET imaging and liquid biopsy with conventional diagnostic tools such as MRI, CT, bone scintigraphy, and serum PSA. This section emphasizes the diagnostic and prognostic advantages of the integrated approach and frames its potential role in different clinical scenarios. To further strengthen the practical value of the review for readers, we also added a new paragraph 4.4. Bridging Clinical Value with Real-World Feasibility that discusses the implementation challenges of PET and liquid biopsy in clinical practice, including aspects such as infrastructure, reimbursement, and standardization. These additions aim to clarify not only the complementary role of the two modalities but also their translational impact and current limitations in real-world settings. We hope that this effort meaningfully enhances the relevance and utility of our review for clinicians and researchers
Round 3
Reviewer 2 Report
Comments and Suggestions for Authors
All the queries has answered, hence my recommendation is "accepted".